# Equilibrium and non-equilibrium furanose selection in the ribose isomerisation network

Avinash Vicholous Dass [1,2], Thomas Georgelin[1,3], Frances Westall[1], Frédéric Foucher [1], Paolo De Los Rios [4,5], Daniel Maria Busiello [4], Shiling Liang [4] & Francesco Piazza [1,6✉]

The exclusive presence of $\beta$-D-ribofuranose in nucleic acids is still a conundrum in prebiotic chemistry, given that pyranose species are substantially more stable at equilibrium. However, a precise characterisation of the relative furanose/pyranose fraction at temperatures higher than about 50 °C is still lacking. Here, we employ a combination of NMR measurements and statistical mechanics modelling to predict a population inversion between furanose and pyranose at equilibrium at high temperatures. More importantly, we show that a steady temperature gradient may steer an open isomerisation network into a non-equilibrium steady state where furanose is boosted beyond the limits set by equilibrium thermodynamics. Moreover, we demonstrate that nonequilibrium selection of furanose is maximum at optimal dissipation, as gauged by the temperature gradient and energy barriers for isomerisation. The predicted optimum is compatible with temperature drops found in hydrothermal vents associated with extremely fresh lava flows on the seafloor.

[1] Centre de Biophysique Moléculaire, CNRS-UPR4301, Rue C. Sadron, Orléans, France. [2] Department of Physics, Ludwig Maximilians University, München, Germany. [3] Laboratoire de Réactivité de Surface, UMR 7197, Sorbonne Université, Paris, France. [4] Institute of Physics, School of Basic Sciences, Ecole Polytechnique Fédérale de Lausanne—EPFL, Lausanne, Switzerland. [5] Institute of Bioengineering, School of Life Sciences, Ecole Polytechnique Fédérale de Lausanne—EPFL, Lausanne, Switzerland. [6] Université d'Orléans, UFR CoST Sciences et Techniques, Orléans, France. ✉email: Francesco.Piazza@cnrs-orleans.fr

Ribose plays a central role in the chemistry of modern life on Earth. The sugar backbones of ribonucleic acids (RNA) and deoxyribonucleic acids (DNA) are formed by polymeric chains of nucleotides which contain a ribose and a deoxyribose (a cyclic pentose without a 2′ oxygen), respectively. Moreover, ribose is a key component in modern metabolism, notably through ATP, which also incorporates its 5-C anomer, i.e. ribofuranose. Therefore, stabilisation and reactivity of ribose formation is a critical topic in prebiotic chemistry and origins of life studies[1].

While the relative stability, abundance, phosphorylation and glycosylation of ribose anomers, and D-ribose ($C_5H_{10}O_5$) in particular, is clearly a pre-eminent issue in the prebiotic scenario of so-called "RNA world"[2,3], it is increasingly being recognised that this topic is, more generally, central to the so-called "metabolism-first" (MF) scenario[4]. As opposed to the "gene-first" scenario, MF hypotheses suggest that life originated from energetically driven geochemical networks that evolved under strong non-equilibrium thermodynamical constraints, eventually leading to the emergence of conserved metabolic pathways at the core of modern biochemical cycles[4]. Of particular relevance in this context appears the role of RNA in scenarios involving collective autocatalytic sets, i.e. autocatalytic ensembles of molecules that can reproduce[5], process and assimilate diversified substrates[6] or, more generally, autocatalytic RNA sets that demonstrate spontaneous self-assembly[7].

As is the case with all carbohydrates, ribose might have been synthesised by polymerisation of formaldehyde in abiotic conditions from the formose reaction[8–10], although this reaction has so far only yielded ribose in very minor quantities[11]. After synthesis, the ribose molecule is unstable in aqueous solution[12]. For example, at pH 7 and 90 °C, its half-life is about 10 h[13]. Despite its lack of stability, ribose is the exclusive constituent of the carbohydrate backbone of RNA in its $\beta$-D-ribofuranose form (hereafter simply furanose). At thermal equilibrium, at room temperature and ambient pressure, this enantiomer represents only 12% of all ribose molecules in solution[14]. Thus, it is of paramount importance to explain the exclusive incorporation of $\beta$-D-ribofuranose in RNA. There are many potential causes for this, ranging from incorporation due to a specific chemical process during phosphorylation or glycosylation, or perhaps due to specific physicochemical conditions on the early Earth that led to a significant increase in the proportion of furanose.

The phosphorylation of ribose could favour a specific enantiomer. It has been shown that the glycosylation of ribose under the $\alpha$-furanose form can lead to the formation of $\beta$-furanose nucleosides[15]. In order to solve the problems of stability and enantiomeric structure, some studies have analysed the ability of ribose to form borate or silicate complexes that are more stable in solution[16,17]. Furthermore, theoretical studies have also shown that silicate/ribose complexes would be formed exclusively from the furanose form because, with this structure, the HO–C–C–OH dihedral angle is sufficiently small to allow the formation of a planar five-membered ring[18]. The silicate or borate scenarios have shown the high potential of inorganic/organic interactions, although the presence of significant borate on the early Earth is unclear[19]. Moreover, since coordination processes seem to have an impact on isomerisation, it is possible that thermal effects at equilibrium could also have an impact on these processes and on the anomeric ratios. Thus, it is important to investigate the effect of temperature on ribose isomerisation at thermal equilibrium. This aspect has not yet been thoroughly studied, despite the fact that temperatures on the primitive Earth, at least at the rock/water interface where prebiotic reactions were taking place, were likely higher than 50 °C[19,20]. It is to be noted, however, that the environments where prebiotic chemical reactions thrived[19–21], such as hydrothermal vents and their immediate vicinity, were by

no means at thermal equilibrium. Most probably, chemical reaction networks proceeded under the action of high, steady gradients of temperature, pH[22] and chemical activity of key molecular species, such as water[19,21,23–26].

It is well known that open chemical systems[27] driven far from equilibrium may settle in non-equilibrium steady states (NESS) that bear little resemblance with equilibrium ones[28]. Well-known examples are regulatory and metabolic networks in biochemistry[29] and, more generally, all chemical transformations that proceed thanks to catalysts, such as enzyme catalysis in biology, where substrate and product species are kept at fixed concentration (chemostatted) by an external source of energy[30]. The work done on the system to enforce the required chemical potential difference leads to sustained dissipative currents that steer the chemical transformations away from equilibrium[31]. In more chemical terms, a NESS should be considered as a regime of sustained kinetic control, with reference to the well-known concept of transient kinetic control of chemical reactions, as opposed to so-called thermodynamic control regime, that is, thermodynamic equilibrium[32]. More recently, the subtle effects of steady temperature gradients on chemical reaction networks have been brought to the fore, especially as regards the coupling of sustained mass currents in physical space and chemical currents in state space, which, based on kinetic rules, may push the steady-state molar fraction of certain chemical species away from equilibrium values[33,34].

In this paper, we focus on a coarse-grained chemical reaction network describing D-ribose isomerisation. Our main working hypothesis is that $\beta$-furanose could be ultra-stabilised under steady non-equilibrium conditions beyond the limits imposed by equilibrium thermodynamics. In order to investigate this idea and quantify the necessary conditions in terms of the unknown kinetic parameters, the first step was to accurately characterise the anomeric ratios at equilibrium as a function of temperature through NMR. The data are fitted to a simple equilibrium model, solved under the constraint of detailed balance. The first finding is that, in view of the large entropic degeneracy associated with sub-conformations of $\alpha$- and $\beta$-furanose, the populations of these high-energy species are predicted to increase with temperature, eventually leading to a population inversion at high temperature.

Equipped with the thermodynamic parameters derived from our experiments, in the second part of the paper we report a theoretical exploration of D-ribose isomerisation under the action of an applied steady gradient of temperature. In this setting, we show that mass currents sustained by the temperature gradient can couple to chemical transformation steps and drive the system into a steady state where the most unstable furanose species can be stabilised beyond the equilibrium limits. The crucial parameters that regulate this effect are (i) energy barriers for chemical transformations and (ii) the Damköhler number, i.e. the non-dimensional ratio between characteristic chemical and mass transport rates.

## Results

**NMR characterisation of D-ribose isomerisation at thermal equilibrium.** In order to investigate the behaviour of ribose isomerisation in solution at equilibrium at increasing temperatures, we carried out $^{13}C$ NMR experiments. This quantitative technique allows unambiguous identification of each anomeric form of ribose by studying the C1 signal of ribose (see Methods). In the following, we denote with $\alpha F$, $\beta F$ and $\alpha P$, $\beta P$ the two enantiomers of furanose (F) and pyranose (P), respectively. We recorded two different sets of spectra. The first set was collected in the temperature range 10–80 °C in pure water. A second set of measurements was conducted in the temperature range 10–25 °C in simulated Hadean sea water[35], with the purpose of investigating isomerisation at equilibrium in the presence of relevant saline

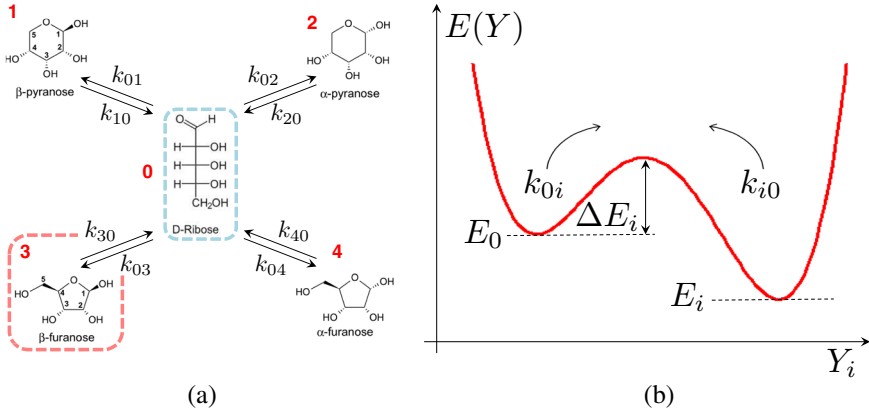

**Fig. 1 Scheme of the D-ribose isomerisation reaction network and model energy landscapes. a** The two pyranoses ($\alpha$ and $\beta$) and the two furanoses ($\alpha$ and $\beta$) are in equilibrium at a given temperature and pressure with the high-energy linear conformation. The rates of chemical transformations, $k_{ij}$, are given by expressions (1). **b** Scheme of the Gibbs free-energy landscape along a representative reaction coordinate for the transition between the linear chain (energy $E_0$) and the $i$-th anomer (energy $E_i$) across the corresponding (unknown) energy barrier $\Delta E_i$. The energy of the linear conformation is the highest, i.e. $E_0 > E_i \; \forall i$[36], corresponding to a population that is undetectable in solution via NMR measurements.

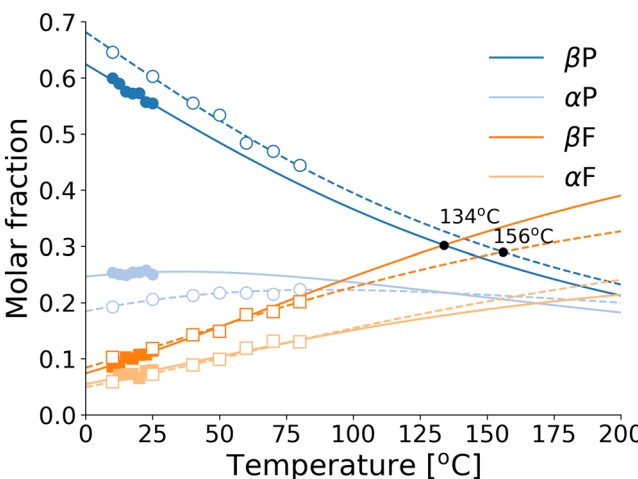

**Fig. 2 Population inversions in D-ribose isomerisation network at thermal equilibrium.** The average molar fractions of the four main anomers of D-ribose are plotted versus temperature for the experiments performed in pure water (open symbols) and in Hadean model water (filled symbols). Lines are fits performed simultaneously with Eq. (2) to the four anomer temperature series (see Table 1). Dashed lines: pure water, solid lines: Hadean model water. In practice, we minimised a single cost function that included a total of $4N_T$ points, where $N_T$ is the number of temperature points considered ($N_T = 7$ for both pure and Hadean water). Error bars (standard errors on the mean) are smaller than the symbol sizes.

conditions. The composition of Hadean oceans was slightly different to that of modern seawater, particularly having lower $SO_4$, being saturated in silica, having higher Fe, Mg, and Ca as well as other elements and molecules related to abundant hydrothermal activity in an ultramafic crust (see Supplementary Note 3 for more information).

The thermal behaviour of D-ribose isomerisation at the resolution of our NMR experiments is governed by the energy (enthalpy) differences between different states, as well as by the degeneracies associated with the different sub-structures of the four main conformations[36], which cannot be resolved in our spectra. Based on these considerations, we can formulate a simple equilibrium model as depicted in Fig. 1 (left panel), that involves the linear conformation and the four ring species, $\beta$P, $\alpha$P, $\beta$F, $\alpha$F. These are labelled with integers from 0 to 4, respectively, as illustrated in Fig. 1. According to the above considerations, the transition rates can be written as the product between a temperature-independent geometric (i.e. entropic) velocity and an Arrhenius-like term, gauging the thermal activation of the transition. Let us denote with $x_i$ ($i = 0, 1,\ldots, 4$) the relative concentration (molar fraction) of the $i$-th species, so that $\sum_{m=0}^{4} x_m = 1$. With reference to Fig. 1 (right panel), we may thus write

$$\begin{cases} k_{0i} = k_{0i}^{\infty} \; e^{-\beta\Delta E_i} \\ k_{i0} = k_{i0}^{\infty} \; e^{-\beta(\Delta E_i + E_0 - E_i)} \end{cases} \quad (1)$$

where $\beta^{-1} = k_B T$, $k_B$ being Boltzmann's constant. The energy of the open linear conformation is the highest, averaging about 19 kJ/mol in the gas phase[36]. This is confirmed by our solution measurements, where this conformation is undetectable. In the limit $\beta E_0 \gg 1$ the stationary solution of the rate equations for the ribose isomerisation network depicted in Fig. 1 at thermal equilibrium read

$$x_i = \frac{\eta_i e^{-\beta E_i}}{\sum_{m=1}^{4} \eta_m e^{-\beta E_m}} \quad i = 1, 2, \ldots, 4 \quad (2)$$

where $\eta_i = k_{0i}^{\infty} / k_{i0}^{\infty}$. The degeneracy factors $\eta_i$ embody the entropic contributions to the interconversion rates, so that the difference in entropy between states $i$ and $j$ can be computed as $\Delta S_{ij} = k_B \log(\eta_i/\eta_j)$. These are mainly associated with the specificities of the multidimensional free-energy landscapes, including local minima corresponding to sub-conformations of each anomer that are in fast equilibrium with the main isoforms corresponding to the four detected NMR lines.

Figure 2 shows clearly that the simple equilibrium model (2) provides an excellent description of the NMR data as temperature increases. The first observation to be made is that the two high-energy furanose species become stabilised as temperature increases, while the equilibrium molar fractions of the more stable pyranoses decrease correspondingly. As a consequence, our analysis predicts that a series of population inversions between the high-energy species $\beta$-furanose and the two low-energy ones, $\alpha$−pyranose, should occur upon increasing the temperature. The first of these should occur at a temperature of about $k_B^{-1}(E_3 - E_2)/\log(\eta_3/\eta_2) \approx 93.2$ °C, while the $\beta$-furanose-$\beta$-pyranose inversion is predicted to occur at a temperature $k_B^{-1}(E_3 - E_1)/\log(\eta_3/\eta_1) \approx 156$ °C. Interestingly, our calculations place the first furanose-pyranose inversion at a temperature some 20 °C lower in Hadean water, at about 134 °C (see Fig. 2).

From a thermodynamic point of view, the population inversions are explicitly related to the large entropic degeneracy associated with the furanose states. The infinite-temperature molar fractions $x_i^\infty$ provide a clear illustration of this trend (even if, of course, the molecular species examined are certainly not stable beyond a certain temperature). From Eq. (2), these read

$$x_i^\infty = \frac{\eta_i}{\sum_{m=1}^{4} \eta_m} \quad i = 1, 2, \ldots, 4 \qquad (3)$$

As illustrated by the calculations reported in Table 1, equilibrium thermodynamics predicts that $\beta$-furanose will dominate at high temperatures. Its stability under given geochemical conditions is thus the only limitation to the maximum fraction of furanose that can be produced at thermal equilibrium by increasing the temperature. Interestingly, this effect is magnified in Hadean water, an environment where temperature-induced boosting of $\beta$-furanose at equilibrium appears to have been easier (see Table 1).

**The furanose population can be boosted beyond equilibrium in a steady temperature gradient.** Our NMR experiments have made it very clear that $\beta$-furanose is progressively stabilised at increasing temperature at thermal equilibrium. However, it is well known that typical prebiotic environments, such as hydrothermal vents and the adjacent porous sediments and chemical precipitates, were characterised by strongly non-equilibrium conditions, such as steady gradients of temperature, pH and water activity[19,37–40]. Intriguingly, in complex systems with multiple states, it has been shown that the rate of dissipation (equivalently, the rate of entropy production) conveys key information on the selection of states that are favoured away from equilibrium[41–43]. In more chemical terms, a given reaction network driven far from equilibrium is placed under a state of sustained kinetic control. In

such conditions, the energy barriers that set the velocity of chemical transformations, which are irrelevant at equilibrium, become key in selecting the steady-state populations.

Taken together, in the context of D-ribose isomerisation kinetics, the above considerations prompt the question whether high-energy furanose species could be further stabilised under steady non-equilibrium conditions, such as an imposed temperature gradient, beyond the limits imposed by equilibrium thermodynamics. The in-depth, quantitative analysis of the thermodynamic equilibrium properties performed in the first part of this study enables us to investigate this question from the vantage point of a model parameterised on solid experimental evidence.

As proposed in a recent work[33], a simple way to examine a chemical reaction network under the action of a steady temperature gradient is to replicate the same system in two separate compartments, each thermalised at a different temperature, and able to exchange reactants as dictated by specific transport rates (see Fig. 3). Furthermore, with no loss of generality, we can coarse-grain the kinetics by letting the two pyranoses coalesce into a single low-energy species, $E_P$, and likewise reunite the two furanoses in a single high-energy moiety, $E_F$. The third species represents the open chain—the high-energy transition state, $E_L$. For simplicity, mass exchange (e.g. convection, diffusion) is described by a single transport rate, $k_D$.

In a system where chemical reactions and transport are coupled, the central physical parameter is the ratio of the respective characteristic rates. If $D$ denotes the typical molecular diffusion coefficient of reactants, $\ell$ some relevant dimension of interest and $\mu$ the typical velocity of chemical transformations (see Fig. 3), then the key parameter is known as the Damköhler number[44], $\mathrm{Da} = \mu\ell^2/D = \mu/k_D$. This parameter measures the

**Table 1 Thermodynamic parameters describing the equilibrium of D-ribose isomerisation in solution at fixed temperature and pressure as estimated by fitting the equilibrium theory (2) to $^{13}$C NMR data.**

| Species | Pure water | | | Hadean water | | |
|---|---|---|---|---|---|---|
| | $E_i$ [kJ/mol] | $\eta_i$ | $x_i^\infty$ | $E_i$ [kJ/mol] | $\eta_i$ | $x_i^\infty$ |
| 1, $\beta$-pyranose | 0 | 0.75 | 0.01 | 0 | 0.39 | 0.01 |
| 2, $\alpha$-pyranose | 6.2 | 3.14 | 0.05 | 4.2 | 0.96 | 0.02 |
| 3, $\beta$-furanose | 13.0 | 29.27 | 0.47 | 14.7 | 29.88 | 0.71 |
| 4, $\alpha$-furanose | 14.2 | 29.41 | 0.47 | 13.1 | 10.88 | 0.26 |

$$k_{\sigma L}(T_2) = \frac{\mu}{\eta_\sigma} e^{-(E_L - E_\sigma + \Delta E_\sigma)/T_2} \qquad k_{\sigma L}(T_1) = \frac{\mu}{\eta_\sigma} e^{-(E_L - E_\sigma + \Delta E_\sigma)/T_1}$$

$$k_{L\sigma}(T_2) = \mu\, e^{-\Delta E_\sigma/T_2} \qquad k_{L\sigma}(T_1) = \mu\, e^{-\Delta E_\sigma/T_1}$$

**Fig. 3 Simple model of ribose isomerisation in a steady temperature gradient ($T_2 > T_1$).** The highest-energy linear species has energy $E_L$, $\alpha$F and $\beta$F have been coalesced into the high-energy state $F$ (energy $E_F$), while $\alpha$P and $\beta$P are coarse-grained into the ground state (and reference energy), $E_P$. Chemical rates are expressed as the product of a velocity, $\mu$, and an Arrhenius term that depends on the energy barriers $\Delta E_\sigma$ ($\sigma = F, P$) and include entropic degeneracy factors, $\eta_\sigma$, in accordance with our experimental findings.

relative time scale of kinetic effects with respect to mass transport and thus offers a single expedient gauge for the degree of coupling between the kinetics of chemical transformations and the transport of reactants and products. In the fast reaction limit, Da ≫ 1, transport is not swift enough to couple molecular species that undergo chemical transformations in separate compartments. Therefore, in the stationary state each box settles at thermal equilibrium at its own temperature.

When transport becomes faster, as happens for example in the presence of strong convective currents in the vicinity of hydrothermal vents[45], exchange of mass between the two compartments will eventually approach timescales comparable to chemical transformations. In this regime, the steady populations of reactants are no longer dictated by detailed balance. Rather, they are governed by a kinetic selection mechanism that depends on the relative values of the energy barriers, $\Delta E_F$, $\Delta E_P$ (see Figs. 1 and 3) and the strength of the temperature gradient, $\Delta T$. Simply put, the molecular species that are generated most quickly from the transition state, $E_L$, will be boosted beyond the equilibrium population by large dissipative mass currents sustained by the temperature gradient. Remarkably, provided the difference between the barriers is large enough, this is true irrespective of whether the fastest state is the least or the most stable at equilibrium. Figure 4a, b illustrate the scenario where the

imposed temperature gradient leads to non-equilibrium boosting of the high-energy furanose species.

Interestingly, while the extent of non-equilibrium stabilisation of furanose in terms of its excess population with respect to equilibrium for $k_D \gg \mu$ (Da ≪ 1) is controlled essentially by the temperature gradient, the crossover to the fast-transport limit, Da < Da*, is also governed by kinetic parameters. This can be encapsulated in a remarkably transparent formula (see Supplementary Note 4), namely

$$\mathrm{Da}^* = \frac{1}{e^{-\Delta E_F/k_B T_M} + e^{-\Delta E_P/k_B T_M}} \qquad (4)$$

where $T_M = (T_1 + T_2)/2 = T_1 + \Delta T/2$ is the average temperature of the system. From Eq. (4) it can be readily seen that the requirement for kinetic selection, i.e. scenarios where one of the barriers is appreciably higher than the average temperature and the other lower, correspond to the timescale-matching condition Da* = $\mathcal{O}(1)$ when the fast reaction is regulated by an energy barrier of roughly the same order as the average temperature.

The rationale behind non-equilibrium stabilisation of furanose becomes obvious once the system of steady fluxes is examined in detail, as illustrated in Fig. 4c (see also Supplementary Note 6). When $E_L \to E_P$ relaxation is the faster pathway out of the

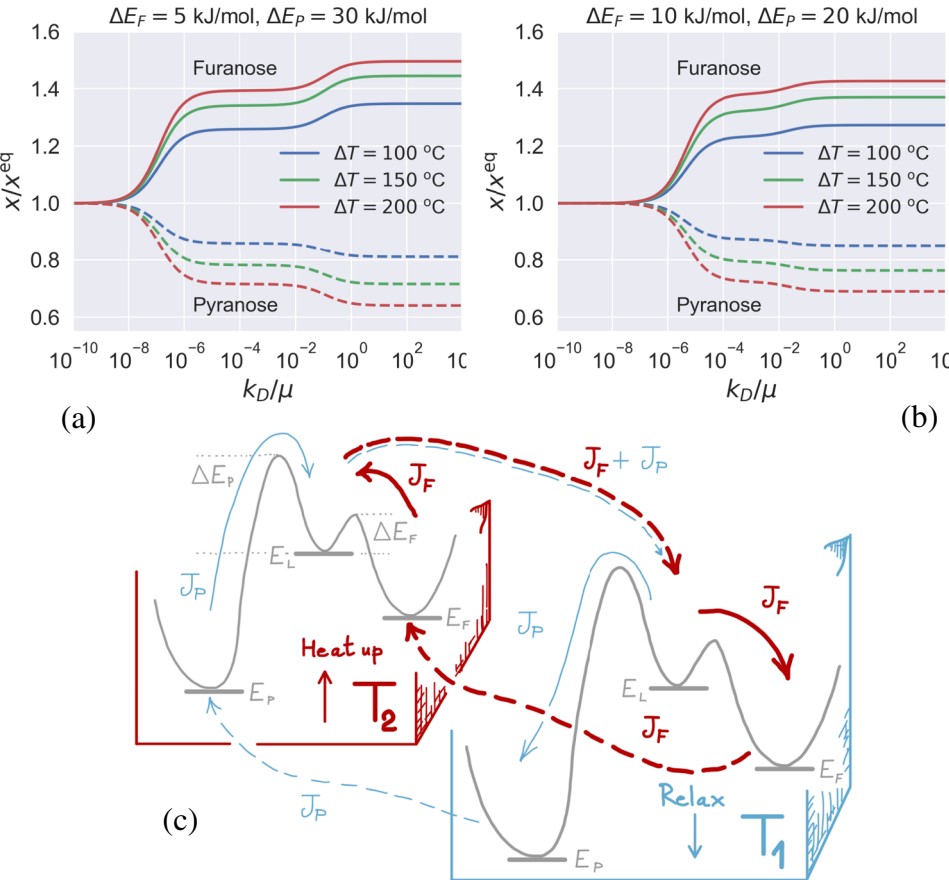

(a)

(b)

(c)

**Fig. 4 Increasing the transport rate for $\Delta E_F < \Delta E_P$ leads to non-equilibrium kinetic selection of furanose beyond thermodynamic equilibrium, as the energy harvested at the hot end is dissipated at the cold end causing population inversion. a, b** The steady-state molar fractions of furanose (solid lines) and pyranose (dashed lines) computed for the two-box model illustrated in Fig. 3 are plotted against the transport rate for different choices of the temperature gradient $\Delta T = T_2 - T_1$ and two choices of energy barriers (see Supplementary Note 4 for the mathematical details). Populations are normalised to the equilibrium values, i.e. $x^{eq} \equiv (x_F^{eq}(T_1) + x_F^{eq}(T_2))/2$. **c** Illustration of the steady system of currents ($J_F$, $J_P$) that sustain the non-equilibrium stabilisation of furanose for $\Delta E_P > \Delta E_F$ (see also Supplementary Note 6). Dashed arrows denote mass transport, solid lines stand for chemical transformations. The current $J_P$ is much smaller than $J_F$ in the fast-transport limit, Da ≲ 1. Parameters used in the two-box model are: $T_1 = 60\,°C$, $\eta_F = 29.34$, $\eta_P = 1.94$, $E_L = 19\,kJ/mol$, $E_F = 13.6\,kJ/mol$, $E_P = 3.1\,kJ/mol$, corresponding to the average values for the two furanose and pyranose enantiomers measured from our equilibrium NMR experiments (see Table 1).

transition state, the furanose population is boosted by a large sustained mass current that supplies high-energy linear molecules from the hot end that undergo ring-closure at the cold side (thick red arrows). A much smaller current circulates in the same direction (thin blue arrows), weakly contributing to the steady mass current of linear D-ribose with pyranose molecules that open up at the hot end.

To summarise, when time scales for transport and chemical transformations match, part of the energy supplied by the temperature gradient can be converted into chemical energy through cycles involving mass transport. Thus, non-equilibrium population inversion may occur if the fastest reaction is fast enough, as gauged by the relative magnitude of $\Delta E_P$ and $\Delta E_F$ (see Fig. 4). The extent of this non-equilibrium effect is illustrated in Table 2, where the temperature needed to obtain a given molar fraction of furanose at equilibrium is compared to the gradient required to obtain the same value in a NESS. The population

boost obtained in a non-equilibrium setting clearly outperforms by far the stabilisation achievable at equilibrium, allowing for molar fractions of furanose that could only be produced at exceedingly high temperatures in thermal equilibrium. Note also that chemical stability might be a less critical issue far from equilibrium, as molecules would fly past the hot end transported by large convective currents.

**Non-equilibrium selection of furanose is maximum at optimal dissipation.** The simple two-box model discussed in the preceding section can be easily generalised in the continuum limit to a system of reaction-diffusion partial differential equations. For the sake of simplicity, and with no loss of generality, we shall restrict ourselves to a one-dimensional system. Let the reactants be confined to a one-dimensional box of length $L$ with reflecting boundary conditions at $x = 0$ and $x = L$ and let $T(x)$ indicate the imposed temperature gradient across the box, or any arbitrary temperature profile with the same boundary conditions, $T(0) = T_1, T(L) = T_2$. We define the space- and time-dependent chemical molar fractions, $\mathcal{P}_\sigma(x, t)$, $\sigma = F, P, L$. The equations then read

$$\frac{\partial \mathcal{P}_F(x, t)}{\partial t} = D\frac{\partial^2 \mathcal{P}_F(x, t)}{\partial x^2} + k_{LF}(x)\mathcal{P}_L(x, t) - k_{FL}(x)\mathcal{P}_F(x, t)$$

$$\frac{\partial \mathcal{P}_P(x, t)}{\partial t} = D\frac{\partial^2 \mathcal{P}_P(x, t)}{\partial x^2} + k_{LP}(x)\mathcal{P}_L(x, t) - k_{PL}(x)\mathcal{P}_P(x, t)$$

$$\frac{\partial \mathcal{P}_L(x, t)}{\partial t} = D\frac{\partial^2 \mathcal{P}_L(x, t)}{\partial x^2} + k_{FL}(x)\mathcal{P}_F(x, t) + k_{PL}(x)\mathcal{P}_P(x, t) +$$
$$- (k_{LF}(x) + k_{LP}(x))\mathcal{P}_L(x, t)$$

$$(5)$$

where $D$ is the diffusion coefficient (assumed to be the same for all species) and the rates are given by the obvious general-

**Table 2 Temperature $T^*$ required to obtain a given overall molar fraction of furanose, $x_F$, at equilibrium, compared to different choices of gradients that yield the same populations in a non-equilibrium steady state.**

|        |       |            | $\Delta E_F = 5$ kJ/mol | | $\Delta E_F = 5$ kJ/mol | |
|        |       |            | $\Delta E_P = 30$ kJ/mol | | $\Delta E_P = 20$ kJ/mol | |
| $T_1$  | $T_2$ | $\Delta T$ | $x_F$ | $T^*$ | $x_F$ | $T^*$ |
|--------|-------|------------|-------|-------|-------|-------|
| 60     | 160   | 100        | 0.47  | 174   | 0.46  | 168   |
| 60     | 210   | 150        | 0.56  | 241   | 0.55  | 234   |
| 60     | 260   | 200        | 0.62  | 301   | 0.61  | 290   |

Two choices of energy barriers are considered. Note that non-equilibrium stabilisation of furanose is more effective in all cases considered, according to the stringent criterion $T^* > T_2$. All temperatures are expressed in °C.

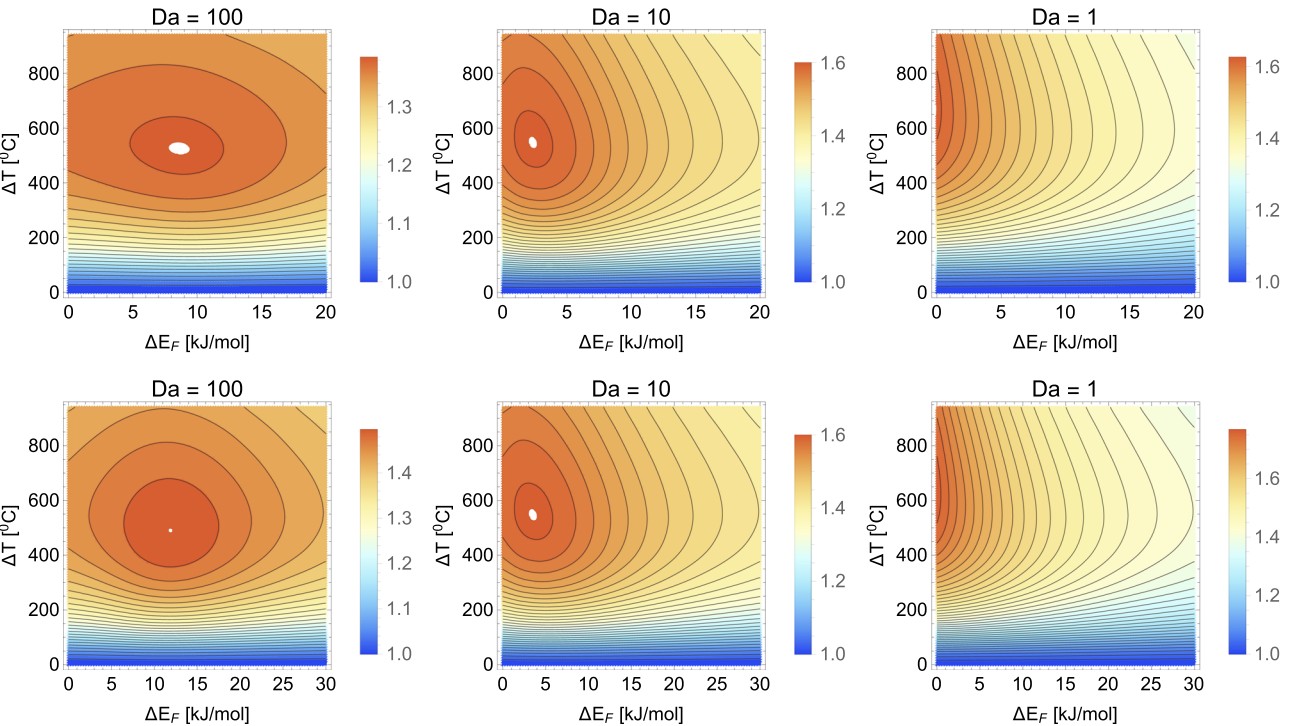

**Fig. 5 Furanose selection is maximum for specific values of the temperature gradient depending on kinetic parameters.** Density plots of the normalised selection indicator $\mathcal{R}_{sel}$, Eq. (7), in the $(\Delta T, \Delta E_F)$ plane computed from the steady-state solutions of the continuum model (5) for different values of the Damköhler number Da $= \mu/DL^2$. The temperature profile is $T(x) = T_1 + \Delta T x/L$. Top panels: $\Delta E_P = 20$ kJ/mol. Bottom panels: $\Delta E_P = 30$ kJ/mol. Other parameters as in Fig. 4.

isations of the expressions introduced in the two-box model (see Fig. 3), that is,

$$k_{FL}(x) = \frac{\mu}{\eta_F} \; e^{-(E_L - E_F + \Delta E_F)/k_B T(x)} \quad k_{LF}(x) = \mu \; e^{-\Delta E_F/k_B T(x)}$$

$$k_{PL}(x) = \frac{\mu}{\eta_P} \; e^{-(E_L - E_P + \Delta E_P)/k_B T(x)} \quad k_{LP}(x) = \mu \; e^{-\Delta E_P/k_B T(x)} \tag{6}$$

The steady-state populations, $\mathcal{P}_\sigma^\infty (x) = \lim_{t\to\infty} \mathcal{P}_\sigma(x,t)$, are the solutions of the system obtained by letting the time derivatives vanish in Eqs. (6), with the normalisation $\mathcal{P}_F^\infty (x) + \mathcal{P}_P^\infty (x) + \mathcal{P}_L^\infty (x) = 1$ (the stationary profiles are nearly flat). In the presence of a temperature gradient, a selection indicator can be defined to quantify the excess fraction of furanose, namely

$$\mathcal{R}_{\text{sel}} = \frac{\int_0^L \mathcal{P}_F^\infty (x) \; dx}{\int_0^L \mathcal{P}_P^\infty (x) \; dx} \frac{\int_0^L \mathcal{P}_P^{\text{eq}}(x) \; dx}{\int_0^L \mathcal{P}_F^{\text{eq}}(x) \; dx} \tag{7}$$

where $\mathcal{P}_\sigma^{\text{eq}}(x) = \eta_\sigma \exp[-E_\sigma/k_B T(x)]/Z$ stands for the equilibrium distributions. By definition, for $T(x) = const.$, the selection indicator (7) is unity.

The plots reported in Fig. 5 not only confirm the results obtained within the two-box model, but also reveal a remarkable fact. Nonequilibrium selection of furanose typically displays a maximum enhancement, corresponding to a restricted region of values of the temperature gradient $\Delta T = T_2 - T_1$ and energy barrier $\Delta E_F$. The maximum appears to correspond to a population boost of 40–50%, for a wide range of transport regimes (as gauged by the Damköhler number) and kinetic parameters, notably the values of the barrier $\Delta E_P$. It can be appreciated that, in the infinite-transport limit, Da < 1, the optimum is progressively pushed to lower values of the energy barrier $\Delta E_F$. Most intriguingly, the optimal value of temperature gradient appears to be consistently not far from the gradients found in hydrothermal venting associated with extremely fresh lava flows on the seafloor, with exit temperatures up to 407 °C recorded[45]. It appears therefore possible that optimal or suboptimal conditions could have been met at certain spots on the bottom of Hadean oceans.

## Discussion

In this paper we have delved into the question of why the β-furanose isoform of D-ribose has been selected as the exclusive sugar component of nucleic acids instead of pyranose, despite being far more unstable and hence present at lower molar fractions in thermodynamic equilibrium up to relatively high temperatures. Our findings suggest that a plausible answer to this question is non-equilibrium enrichment beyond equilibrium sustained by a temperature gradient.

In the first part of this work, we analyse the results of NMR measurements of ribose isomerisation at equilibrium at increasing temperatures. A simple thermodynamic equilibrium model reproduces the NMR data to an excellent extent, revealing that the populations of the more unstable furanose species increase with temperature, while those of pyranoses decrease. As a result, a population inversion to a furanose-richer phase is predicted to occur at a temperature of about 150 °C, directly connected to the large entropic degeneracy associated with furanose internal isoforms. Remarkably, the same measurements repeated in the presence of a salt mixture simulating that of Hadean oceans led to an inversion temperature about 20 °C lower.

While high temperatures must have been ubiquitous in the early Earth environment, at least close to the rock/water interface, it is highly unlikely that the kind of hotbeds suggested as possible primeval chemical reactors were at thermodynamic equilibrium. Building on this idea, and on the physical parameters measured in our NMR experiments, in the second part of the paper we pursue the idea that furanose might have been stabilised beyond the

limits imposed by equilibrium thermodynamics under non-equilibrium conditions. To this end, we investigate a simple model of ribose isomerisation in the presence of a steady temperature gradient. Our calculations show that driving the network far from equilibrium may lead to sustained kinetic selection of furanose, i.e. increased stabilisation with respect to thermal equilibrium, if linear-to-furanose interconversion proceeds appreciably faster than linear-to-pyranose.

The increased stabilisation occurs as diffusive and convective mass currents set in, shuttling molecules cyclically across the temperature gradient. Thus, in the NESS, sustained mass transport and chemical transformations may couple in such a way that furanose molecules absorb heat at the hot side and are transported to the cold end, where the extra heat is used to steer chemical relaxation towards the production of more furanose. We show that this scenario may emerge provided (i) the typical Damköhler number of the system is lower than about 1–10 (depending on energy barriers), and (ii) the relaxation of linear ribose to furanose is faster than its relaxation to pyranose. According to the first requirement, timescales characteristic of mass transport should roughly match those typical of chemical transformations. If this is the case, the fastest reactions will be kinetically selected, and the populations of associated products boosted beyond thermal equilibrium through sustained mass currents. The second requirement amounts to a condition on the relative magnitude of the energy barriers that separate chemical states in the system. While these are strictly irrelevant at equilibrium, in systems driven away from equilibrium, (kinetic) selection of the fittest may be achieved as selection of the fastest[33,46]. More generally, in non-equilibrium conditions, the fitness associated with a given energy landscape can be thought of as being shifted kinetically[47,48].

Variations of the temperature gradient and energy barriers have a profound effect on the extent of furanose non-equilibrium selection. We show that ultra-stabilisation is maximum in specific regions of parameter space. Intriguingly, optimal temperature gradients are predicted in the range 300–400 °C, of the same order of magnitude as those found in the proximity of present-day venting activity at some spots on the seafloor[45]. This prompts the intriguing hypothesis that kinetic landscapes such as those illustrated in Fig. 5 might have been integrated by evolution in more comprehensive, multi-fitness landscapes that eventually led to the biochemistry of life that we know today.

In summary, an accurate thermodynamic characterisation of ribose isomerisation revealed the occurrence of population inversion between furanose and pyranose species at increasing temperatures in thermal equilibrium. However, we demonstrated that the constraints imposed by equilibrium thermodynamics can be overcome and more furanose produced by driving the network far from equilibrium through a steady temperature gradient.

It should be recognised that alternative possibilities that do not necessarily require a pre-existing abundance of ribofuranose for its selection have been proposed. In a more systemic perspective, these should be considered at least as concurrent selection pathways. Historically, the Formose reaction has been considered the major pathway, by which ribose could be formed. However, detailed studies revealed the formation of branched side products, low yields of ribose and high level of degradation that paved the way for criticism[12]. The formation of ribose bisphosphate from glycolaldehyde phosphate and formaldehyde, with similar starting materials as the formose reaction, is suggested as an alternative pathway by which ribose could have been driven to be a major product[49]. But then again, the degradation kinetics of ribose and ribose bisphosphate have always been of concern and one of the prime reasons for ribose being dubbed a non-ideal candidate for the origin of RNA as genetic material[12]. It was suggested that

ribose might have been conserved in contemporary metabolic pathways, such as the pentose pathway (and subsequently into the glycolysis pathway), by kinetically controlled reaction of ribose (e.g. from formose reaction) with HCN, forming stable lactone and aldonic acid which were later converted back to ribose after the onset of enzymes[12]. More recently, Eschenmoser suggested the necessity to revisit the HCN chemistry and suggested a Glyoxlate scenario of primordial metabolism[50]. Subsequent studies on the reaction of dihydroxyfumarate (with glyceraldehyde) have shown clean reactions where $\alpha$ and $\beta$ ribulofuranoses dominate[51]. Such a reaction could potentially integrate sugars into the modern metabolic pathways.

The abiotic origin of DNA is still uncertain and it is unclear if the deoxynucleic acids emerged independently or as a product of biotic selection. Our current study is limited to ribose and its anomers. Nevertheless, within a non-equilibrium thermal gradient setting, a similar argument in the case of the deoxyribofuranose might not be unreasonable, given the fact that the 2-deoxyribose exhibits a similar anomeric distribution as ribose, i.e. the deoxypyranoses dominate at equilibrium[52] up to 36 °C. Thus, we expect 2-deoxyribofuranoses could also be enriched using our model. Nonetheless, such a result would also only account for one of the pathways by which $\beta$-deoxyfuranoses might have been utilised as a component of nucleic acids. However, alternate routes to the synthesis of 2-deoxyribofuranosides using photoredox chemistry exist that do not necessitate an enrichment mechanism[53]. Recently, it has been experimentally demonstrated that an aldol reaction between acetaldehyde and glyceraldehyde, promoted by amino nitriles, leads to the formation of 2-deoxyribose[54] with $\simeq 5\%$ yields. In such a scenario, where a concentration mechanism is necessitated in order to accumulate molecules, our non-equilibrium model would be advantageous.

It should be stressed that our study does not necessarily preclude different means by which $\beta$-ribofuranoses or even ribose might have been selected as the principal component of nucleic acids. Eschenmoser elegantly reasoned the various possibilities and limitations of other aldoses and their corresponding nucleic acids as prime candidates with potential to be genetic polymers[55]. He suggested that alternative sugar modules such as hexopyranoses could have competed with ribose, but might have been left behind in the prebiotic competition due to their inherent inability to act as informational carriers of genetic information (due to weak base pairing properties). The base pairing properties of these alternate nucleic acids further faded when the deoxyhexopyranoses were substituted as the sugar units. The lack of functional prerequisite is not the only cause for evolutionary rejection, but excessive stability of the base pairing in duplex state could also lead to an evolutionary exclusion strategy. Such is the case of pentopyranosyls nucleic acids, which form extremely stable base pairs[55].

Our study in its present state of development is restricted to investigating the influence of thermal equilibrium and non-equilibrium conditions on the relative anomeric ratios in a D-ribose enriched system. It cannot be neglected that the Hadean Earth was far from such a model system. Factors like pH, loss of molecules due to dilution, presence of various minerals, reactivity and degradation would affect this isomerisation phenomenon to varying extents. Concerning pH or other non-thermal effects, it should be noted that, unless pH (or other factors) stabilises ribofuranose intrinsically, a pH gradient would also be effective as long as the barriers of the transitions between the states are modulated by local pH, as they are by temperature.

Sugars have been successfully synthesised under alkaline, mineral-assisted conditions. Minerals like borates[56] and silicates[18,57,58] are known to enhance the formation of pentoses. In fact, the most efficient ways in which such minerals are able to stabilise these sugars are with the furanoses that complex with these minerals due to the favourable dihedral angle of their hydroxyl groups. It is also important to note that the degradation pathways of the sugars are mainly via the open-chain[59] and the pyranose forms[60], which are more reactive in comparison to the furanoses, thus supporting the fact that the ribofuranoses could also serve as a shielding mechanism against the degradation of sugars and in turn enhance their concentration and promote complexation with minerals like borates and silicates (the latter especially known to be abundant on the early Earth). With respect to the reactivity of these sugars, it has been demonstrated that a borate-complexed ribose is regioselectively phosphorylated under dry conditions[56] and in a biphasic system of aqueous formamide[61]. More recently, under non-equilibrium thermal conditions, phosphorylation of nucleoside has been demonstrated successfully[62].

In summary, our study should be considered as an attempt to quantify one among several possibilities through which the emergence of contemporary nucleic acid component $\beta$-ribofuranose would be favoured. In this regard, our results demonstrate the subtle non-equilibrium physicochemical effects that may arise from the interplay of vast chemical landscapes and the geophysical conditions of their surroundings on primordial Earth.

## Methods

### NMR measurements

*Choice of the relaxation time* $T_1$ *in the NMR experiments.* All NMR spectra were collected at ambient pressure in capped tubes. Checks with sealed tubes were made to ensure that the results did not change. The first step in order to quantify the relative population of each anomer through $^{13}C$ NMR experiments is to evaluate the relaxation time $T_1$ for all species. Since $T_1$ is strongly dependent on temperature, $T_1$ measurements were carried out at each temperature before the spectral data were recorded. The results of these experiments are reported in Supplementary Table 1. $T_1$ was found to increase significantly with temperature from about 2–5 s, for temperatures between 10 and 80 °C, the temperature range of this study. Overall, however, the values of $T_1$ are similar for all anomers at a given temperature. Only $\alpha$-pyranose shows a significantly higher value at the higher temperature of 80 °C. Building on these results, we investigated the anomerisation of ribose by $^{13}C$ NMR at different temperatures, by adapting the $D_1$ ($5T_1$) for each experiment.

*Extracting the equilibrium molar fractions from the NMR spectra.* At 25 °C ($T_1 = 1.8$ s), the signals of C1-$\alpha$P, C1-$\beta$P, C1-$\alpha$F and C1-$\beta$F are found, respectively, at 93.56, 93.85, 96.33, and 100.99 ppm. A large number of $^{13}C$ NMR spectra, between 50 and 80 °C, were recorded for each temperature and the average relative molar fractions computed by fitting the corresponding peak areas with Lorentzian line shapes. This procedure is illustrated for one representative spectrum in Supplementary Fig. 1.

## Data availability

The datasets generated and/or analysed during the current study are available from the corresponding author on reasonable request.

## Code availability

The codes used in this study are available from the corresponding author on reasonable request.

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

## Acknowledgements

A.V.D. and F.W. would like to acknowledge Region Centre, France, for a doctoral bursary for A.V.D. P.D.L.R. and S.L. thank the Swiss National Science Foundation for support under grant 200020_178763.

## Author contributions

F.P., T.G., A.V.D. and F.W. designed the study. A.V.D. and T.G. carried out the NMR experiments. F.P. worked out the equilibrium model and performed the associated calculations. F.P., P.D.L.R. and D.M.B. worked out the non-equilibrium model. F.P. and D. M.B. performed the numerical and analytical calculations for the latter model. F.P. wrote the first draft of the manuscript. F.P., F.W., T.G., A.V.D., F.F., P.D.L.R., D.M.B., S.L. read, commented and corrected the subsequent revisions. A.V.D. supplied the additional discussion in the last part of the Discussion section.

## Competing interests

The authors declare no competing interests.
