## [Peer Review File · Nature Communications]

REVIEWER COMMENTS

Reviewer #1 (Remarks to the Author):

Dear Editor

the study presented by Dr. Piazza and collaborators deals with the characterization of the D-ribose alpha/beta-furanose vs alpha/beta-pyranose equilibrium at different temperature and with the possible non-equilibrium states derived from temperature gradients.

The Authors successfully apply NMR to characterize the equilibrium reporting that the furanose isomer becomes stable at high T, while the pyranose one is the dominant isomer at low T. Starting from this basic information, the Authors further studied, by means of calculations, the scenario whereby a T gradient would drive (in non-equilibrium conditions) an inversion of the populations between the different isomers, taking into account the ratio between reaction rates and diffusion rates.

With respect to technical issues, I have no objections to the study.

Instead my major comment will refer to the whole vision here proposed, and in particular to the very central idea that because the furanose isomer is today present in RNA, this means that primitive conditions must assure that furanose becomes the 'selected' molecule. In other words, the underlying idea of the Authors is that in order to justify the existence of RNA, one should firstly demonstrate that ribofuranose is selected among other isomers. From this reasoning, it stems that one should find a physico-chemical motivation for letting the ribofuranose emergence (or at least for increasing its abundance).

The corresponding linear causality (RNA emerges because ribofuranose was more abundant) can be only one of the possible explanations.

In order to do so, the Authors have created a highly idealized world, where ribose is "alone" and the fates of its isomers are determined only by thermodynamic-kinetic features. No surfaces, no other molecules, no other chemical or physical processes concur to the determination of what is happening. This scenario appears quite unrealistic and especially does not take into account the possible role that other molecules could play in a complex primitive scenario. Also, the central idea refers to an abiotic origin of ribose (via the formose reaction) that would place the formation of

sugars and nucleotides (and RNA) as independent and precedent the origin of many other molecules. The latter, however, could affect the formation of ribofuranose in chemical ways, by binding, catalysis, by supramolecular selection.

The very simplified scenario of a naked ribofuranose/ribopyranose system and its thermodynamic/kinetic features appears an idealized system that can be published, of course, but it should be properly placed in a realistic context.

I see that such kind of discussion is now absent in the manuscript, and thus my suggestion is to present the system under investigation with proper caution. This remark does not aim at underrating the scientific rigour and accuracy of the study, but would like to bring the Authors toward a more inclusive view of different possible scenario. The a posteriori selection of ribofuranose, for example, could be justified by features of RNA (as we know it today) that would not be present in a hypothetic ribopyranose RNA. In this respect, the studies on aetiology of biopolymers done by Eschenmoser could be included in the discussion.

The text would be very much improved if the Authors recognize the limitation of thinking in terms of a highly idealized isolated-ribose world in favour of a more systemic perspective, including – at least in the discussion – alternative possibilities that does not require necessarily a pre-existing abundance of ribofuranose for its selection (selection can occurs 'downward', i.e., structures at higher complexity that select for their building block, and not the reverse; or can occurs due to multimolecular interaction – and especially supramolecular binding and catalysis, whereby the initial abundance of a certain isomer is not a prerequisite for selection).

With respect to minor issues, please note: on page 13, 34% vs 50% does not appear a critical difference; in Table 2: is 260 degrees Celsius a realistic temperature for chemical stability?

What if the entire model is complemented by a degradation reaction? Does it still work as currently?

Reviewer #2 (Remarks to the Author):

This paper presents an important analysis for the origin of life and fundamental organic chemistry, investigating the dynamics between different isoforms of ribose and potential physicochemical conditions that could have caused the selection of beta-furanose instead of pyranose in biochemistry (even though the latter is more stable in thermodynamic equilibrium). It is well written and clear, and has implications for different theories on geochemical settings for abiogenesis, particularly by highlighting the effects of non-equilibrium in chemical transformations. It is timely and relevant, and the conclusions are supported by the results. My comments are mostly related to clarification, completion, and presentation to a broader audience.

1. In the Abstract the authors mention nucleic acids, in the Introduction RNA only; please start from nucleic acids also in the Introduction, describing briefly to a broader audience the difference between the rings in RNA and DNA. In the Discussion, the authors could also bring back deoxyriboses in one sentence, and predict if they think their results would be similar if using those, and if they think deoxyriboses need come after the selection that they demonstrate, which has implications for the order of appearance of DNA vs RNA.

2. In the first sentence of the Introduction, the authors do a disservice to themselves in highlighting the RNA World (particularly with an old citation). A broader statement that highlights the relevance of their work to multiple prebiotic scenarios would be more inclusive and attract a broader readership. This is, RNA figures in other scenarios than the traditional formulation of the RNA World as cited, e.g. in recent work demonstrating autocatalytic networks of RNA (see Vaidya et al. *Nature* 491, 72–77(2012)). But beyond that, other molecules include the furanose ring, noteworthy the ATP molecule itself. Therefore, this work is relevant to theories usually under the umbrella of "Metabolism First" as well, and would benefit from agnosticism in this regard.

3. Please explain the Damköhler number in the Introduction the first time it is mentioned, even if briefly.

4. The authors and editor should reconsider the amount of Methods/Results/Discussion that is presented in the end of the Introduction. A brief description of the main results is welcomed, but in my view at present that part extends beyond the expected, with almost a full page.

5. In the beginning of the results (a "Results" heading would be helpful, if the journal allows) please describe/specify what is "simulated Hadean sea water", even if briefly (this is well-described in the Supplementary Material but please find one sentence or two for this location, for better contextualization)

6. The placement of the molecular structures of the riboses in Figure 3 is very sub-optimal. Reading the caption one understands that they correspond to each state in the boxes, but with the current placement in the figure that is not evident at all, they just seem to be there, particularly alpha-P and Beta-P are not aligned at all with the ground state EP. Please make the figure clearer with a better alignment of the structures with each of the energetic states, and perhaps some faint grey horizontal lines to make the association more clear, perhaps removing T1 and T2 from the boxes to the top, allowing EP to be lower.

7. This work focus on temperature and I believe the experiments and modeling are sufficient to draw the conclusions presented, but the Discussion would be significantly enriched with a small paragraph exploring the predicted effects of pH in the results. What would be the effect of pH gradients in this scenario? What would be the effect of high alkalinity, prominent in some hydrothermal vent scenarios that are supported by the results of this work?

8. This is blatant and I am not sure how it was missed between 8 authors, but the reference numbers are placed after punctuation, both after commas and full stops/periods. Reference numbers should be placed before the punctuation, within the sentence/fragment that they apply to.

Minor

1. Lines are not numbered, please do so next time.

2. There are problems with references, e.g. ref. nr 4. was published in PNAS and the journal name is another, strange one. Please double-check the reference list

3. Consider using "catalyst" instead of "catalyser"

4. grammar: "In this paper, we focus on a coarse-grained chemical reaction networks"

5. Figure 1, Figure 4- please use a) and b), instead of "left", "right" and "top", "bottom", placed accordingly in the figure, referenced properly in the text to e.g. Fig.1a instead of Fig. 1 (right panel)

6. consider removing unnecessary statements as "It is not difficult to check that," and "It turns out that", they are distracting.

7. page 9, PH should read pH, velocit should ready velocity or speed

8. please remove "see again" when you refer to a figure for a second or third time. Just the figure number is sufficient, this is distracting.

9. page 17, bottom "is of lower"

10. There are some weird commas (separating subject from verb) throughout the text, please double-check. E.g "The second requirement, amounts to"

11. On my end, Figure S5, S6 and S8(Top) do not show/the pdf is corrupted.

12. Consider digitizing the hand-drawn Figure 4 bottom.

Reviewer #3 (Remarks to the Author):

In this interesting work, the authors provided solid evidence that the nonequilibrium temperature gradient is crucial for the kinetic furanose selection in the ribose isomerisation network. I am enthusiastic about the publication of this work after addressing some comments below:

1. It is important to realize that the nonequilibrium states including the steady state are characterized by both the landscape and the temperature gradient which is very different from the equilibrium where the landscape barrier itself is enough. See Proc. Natl. Acad. Sci. USA , 105: 12271-12276. (2008); Advances in Physics, 64:1, 1-137. (2015); Reviews of Modern Physics 91(4), 045004 (2019).

2. The kinetic effect of the temperature gradient is a curl, or rotational flow driving the motion in state space as authors envisioned as the cyclic motion. This has a rigorous mathematical foundation and can be quantified physically, as shown in Proc. Natl. Acad. Sci. USA , 105: 12271-12276. (2008); Advances in Physics, 64:1, 1-137. (2015); Reviews of Modern Physics 91(4), 045004 (2019).

3. The effect of the nonequilibrium effect through the temperature gradient or the equivalent rotational driving force is to shift the fitness. There is a mathematical foundation behind this. See reference J. Chem. Phys., 137, 065102. (2012) to distinguish the evolution fitness and the population probability in the evolution. Under the equilibrium condition or Wright's studied simple cases, they are equal. However, when the nonequilibrium effects are important, the original fitness is no longer the optimal of the population any more.

Point-by-point response to the reviewers' comments

Reviewer #1

The study presented by Dr. Piazza and collaborators deals with the characterization of the D-ribose alpha/beta-furanose vs alpha/beta-pyranose equilibrium at different temperature and with the possible non-equilibrium states derived from temperature gradients.

The Authors successfully apply NMR to characterize the equilibrium reporting that the furanose isomer becomes stable at high T, while the pyranose one is the dominant isomer at low T. Starting from this basic information, the Authors further studied, by means of calculations, the scenario whereby a T gradient would drive (in non-equilibrium conditions) an inversion of the populations between the different isomers, taking into account the ratio between reaction rates and diffusion rates.

With respect to technical issues, I have no objections to the study.

Instead my major comment will refer to the whole vision here proposed, and in particular to the very central idea that because the furanose isomer is today present in RNA, this means that primitive conditions must assure that furanose becomes the 'selected' molecule. In other words, the underlying idea of the Authors is that in order to justify the existence of RNA, one should firstly demonstrate that ribofuranose is selected among other isomers. From this reasoning, it stems that one should find a physico-chemical motivation for letting the ribofuranose emergence (or at least for increasing its abundance).

The corresponding linear causality (RNA emerges because ribofuranose was more abundant) can be only one of the possible explanations.

In order to do so, the Authors have created a highly idealized world, where ribose is "alone" and the fates of its isomers are determined only by thermodynamic-kinetic features. No surfaces, no other molecules, no other chemical or physical processes concur to the determination of what is happening. This scenario appears quite unrealistic and especially does not take into account the possible role that other molecules could play in a complex primitive scenario. Also, the central idea refers to an abiotic origin of ribose (via the formose reaction) that would place the formation of sugars and nucleotides (and RNA) as independent and precedent the origin of many other molecules. The latter, however, could affect the formation of ribofuranose in chemical ways, by binding, catalysis, by supramolecular selection.

The very simplified scenario of a naked ribofuranose/ribopyranose system and its thermodynamic/kinetic features appears an idealized system that can be published, of course, but it should be properly placed in a realistic context.

I see that such kind of discussion is now absent in the manuscript, and thus my suggestion is to present the system under investigation with proper caution. This remark does not aim at underrating the scientific rigour and accuracy of the study, but would like to bring the Authors toward a more inclusive view of different possible scenarios. The a posteriori selection of ribofuranose, for example, could be justified by features of RNA (as we know it today) that would not be present in a hypothetical ribopyranose RNA. In this respect, the studies on aetiology of biopolymers done by Eschenmoser could be included in the discussion.

The text would be very much improved if the Authors recognize the limitation of thinking in terms of a highly idealized isolated-ribose world in favour of a more systemic perspective, including – at least in the discussion – alternative possibilities that does not require necessarily a pre-existing abundance of ribofuranose for its selection (selection can occur 'downward', i.e., structures at higher complexity that select for their building block, and not the reverse; or can occur due to multimolecular interaction – and especially supramolecular binding and catalysis, whereby the initial abundance of a certain isomer is not a prerequisite for selection).

We thank the referee for their appreciation of our work and for the deep and thoughtful comments about the lack of a somewhat broader perspective in the contextualization of our study. We have expanded the text (introduction and conclusions) along the lines suggested by the referee, notably to tone down the main idea that the initial abundance of a certain isomer is the main prerequisite for selection and inscribe this hypothesis in a

broader picture. In particular, we have introduced in the revised text (section “Summary and discussion”) a thorough discussion of our ideas in the context of the studies on aetiology of biopolymers performed by Eschenmoser.

Of course, the referee is correct in pointing out conceptual routes to selection of a certain molecule that are alternative to mere overabundance within a certain network of reactions. Nevertheless, from a sheer probabilistic point of view, it is legitimate to admit that the more abundant a molecule, the more it has a chance to be employed as a substrate (or catalyst) in specific reaction pathways. After all, life as we know it is mostly based on the most abundant elements in the Universe. In other words, the fact that ribofuranose could have been more abundant than expected before in hydrothermal environments due to thermal gradients may have favored its use and the formation of RNA. Of course, this could have been one among many, possibly concurrent selection routes. More generally, it is worth noting that the same logic is applied in studies related to homochirality, where researchers investigate conditions leading to enantiomeric excess in order to explain the use by life of L or D molecules.

With respect to minor issues, please note: on page 13, 34% vs 50% does not appear a critical difference;

We agree with the referee. We have toned down the comment on this difference in the revised manuscript. Nevertheless, it should be observed that this comparison in the text is meant as an introduction to the calculations reported in Table 2, aimed at illustrating the nonequilibrium effect associated with a given temperature gradient as compared to the enrichment in ribofuranose obtained at equilibrium. By contrast, the figures reported in Table 2 highlight a much stronger effect associated with the nonequilibrium setting.

in Table 2: is 260 degrees Celsius a realistic temperature for chemical stability?

It probably is not. That is part of the interest associated with the nonequilibrium scenario. Molecules would have to be transported rapidly past the hot source at 260 °C by convective currents that circulate between the cooler and hotter parts of the hydrothermal system to produce a furanose enrichment that would necessitate unrealistic temperatures (in terms of chemical stability) at equilibrium.

What if the entire model is complemented by a degradation reaction? Does it still work as currently?

This is an interesting point. Our system is closed, i.e. mass is conserved. In the presence of a degradation reaction and in the absence of sources, one would observe the same effect as a *transient*, provided the degradation rate is sufficiently low with respect to the characteristic rate of mass transport across the gradient. If, however, a source term is also included, the system would select a different nonequilibrium stationary state, which will depend on several factors, notably (i) which species are supplied and degraded, the rates of injection (synthesis) and degradation and, possibly more interestingly, (iii) the spatial pattern of the mass sources and sinks.

Reviewer #2

This paper presents an important analysis for the origin of life and fundamental organic chemistry, investigating the dynamics between different isoforms of ribose and potential physicochemical conditions that could have caused the selection of beta-furanose instead of pyranose in biochemistry (even though the latter is more stable in thermodynamic equilibrium). It is well written and clear, and has implications for different theories on geochemical settings for abiogenesis, particularly by highlighting the effects of non-equilibrium in chemical transformations. It is timely and relevant, and the conclusions are supported by the results. My comments are mostly related to clarification, completion, and presentation to a broader audience.

We thank the referee for their highly appreciative remarks. Please find below point-to-point replies and list of changes made to the revised manuscript accordingly.

1. In the Abstract the authors mention nucleic acids, in the Introduction RNA only; please start from nucleic acids also in the Introduction, describing briefly to a broader audience the difference between the rings in RNA and DNA. In the Discussion, the authors could also bring back deoxyriboses in one sentence, and predict if they think their results would be similar if using those, and if they think deoxyriboses need come after the selection that they demonstrate, which has implications for the order of appearance of DNA vs RNA.

The referee makes a good point here. Our current study is limited to ribose and its anomers. Nevertheless, within a non-equilibrium thermal gradient setting, a similar argument in the case of the deoxyribofuranose might not be unreasonable, given the fact that the 2-deoxyribose exhibits a similar anomeric distribution as ribose, i.e. the deoxyfuranoses dominate at equilibrium¹ up to 36 °C. Thus, we expect 2-deoxyribofuranoses could also be enriched using our model. Nonetheless, such a result would also only account for one of the pathways by which β -deoxyfuranoses might have been utilised as a component of nucleic acids. However, it should be noted that alternate routes to the synthesis of 2-deoxyribofuranosides using photoredox chemistry exist that do not necessitate an enrichment mechanism².

We have added a brief description of the difference between the rings in RNA and DNA in the introduction as requested. Moreover, we have considerably expanded the discussion in the final section along these lines in order to discuss our results in terms of the appearance of deoxyribose.

2. In the first sentence of the Introduction, the authors do a disservice to themselves in highlighting the RNA World (particularly with an old citation). A broader statement that highlights the relevance of their work to multiple prebiotic scenarios would be more inclusive and attract a broader readership. This is, RNA figures in other scenarios than the traditional formulation of the RNA World as cited, e.g. in recent work demonstrating autocatalytic networks of RNA (see Vaidya et al. Nature 491, 72–77(2012)). But beyond that, other molecules include the furanose ring, noteworthy the ATP molecule itself. Therefore, this work is relevant to theories usually under the umbrella of "Metabolism First" as well, and would benefit from agnosticism in this regard.

The referee is absolutely right. We have expanded the introduction in the revised manuscript so as to inscribe our work in the broader picture of “metabolism-first” hypotheses beyond the “RNA-world” citation. Moreover, we have added the following more recent citation to the RNA world:

Higgs, P., Lehman, N. The RNA World: molecular cooperation at the origins of life. *Nat Rev Genet* **16**, 7–17 (2015). <https://doi.org/10.1038/nrg3841>

3. Please explain the Damköhler number in the Introduction the first time it is mentioned, even if briefly.

Done as instructed.

4. The authors and editor should reconsider the amount of Methods/Results/Discussion that is presented in the end of the Introduction. A brief description of the main results is welcomed, but in my view at present that part extends beyond the expected, with almost a full page.

We have trimmed this part down to about half a page.

5. In the beginning of the results (a "Results" heading would be helpful, if the journal allows) please describe/specify what is "simulated Hadean sea water", even if briefly (this is well-described in the Supplementary Material but please find one sentence or two for this location, for better contextualization)

Concerning the naming strategy for sections, we have opted for slightly more detailed headlines, providing a concise glimpse of the material to follow. We would rather stick to that choice, but can revert to more general sectioning as suggested by the referee if that is required by the journal policy.

We have added one new sentence when Hadean water is first mentioned (page 6), which relates the essential of the water model that we used in anticipation of the more comprehensive description reported in the supplementary material.

¹ Cortes, Sergio J, Robert L Van Etten, and Tony L Mega. 1991. “The 18O Isotope Shift in 13C Nuclear Magnetic Resonance Spectroscopy. 14. Kinetics of Oxygen Exchange at the Anomeric Carbon of D-Ribose and D-2-Deoxyribose.” *Journal of Organic Chemistry* 56 (3): 943–47. <https://doi.org/10.1021/jo00003a009>.

² Xu, Jianfeng, Nicholas J. Green, Clémentine Gibard, Ramanarayanan Krishnamurthy, and John D. Sutherland. 2019. “Prebiotic Phosphorylation of 2-Thiouridine Provides Either Nucleotides or DNA Building Blocks via Photoreduction.” *Nature Chemistry* 11 (5): 457–62. <https://doi.org/10.1038/s41557-019-0225-x>.

6. The placement of the molecular structures of the riboses in Figure 3 is very sub-optimal. Reading the caption one understands that they correspond to each state in the boxes, but with the current placement in the figure that is not evident at all, they just seem to be there, particularly alpha-P and Beta-P are not aligned at all with the ground state EP. Please make the figure clearer with a better alignment of the structures with each of the energetic states, and perhaps some faint grey horizontal lines to make the association more clear, perhaps removing T1 and T2 from the boxes to the top, allowing EP to be lower.

We understand the referee's concerns about the readability of this figure. We have modified it in the revised version in the attempt of making it more clear by implementing the referee's suggestions. In fact, we believe that the figure has improved substantially. We include it here to facilitate the task of checking.

$$k_{\sigma L}(T_2) = \frac{\mu}{\eta_{\sigma}} e^{-(E_L - E_{\sigma} + \Delta E_{\sigma})/T_2} \quad k_{\sigma L}(T_1) = \frac{\mu}{\eta_{\sigma}} e^{-(E_L - E_{\sigma} + \Delta E_{\sigma})/T_1}$$

$$k_{L\sigma}(T_2) = \mu e^{-\Delta E_{\sigma}/T_2} \quad k_{L\sigma}(T_1) = \mu e^{-\Delta E_{\sigma}/T_1}$$

7. This work focus on temperature and I believe the experiments and modeling are sufficient to draw the conclusions presented, but the Discussion would be significantly enriched with a small paragraph exploring the predicted effects of pH in the results. What would be the effect of pH gradients in this scenario? What would be the effect of high alkalinity, prominent in some hydrothermal vent scenarios that are supported by the results of this work?

This is an important point indeed. Concerning pH, or other non-thermal effects: unless pH (or others) intrinsically stabilizes ribofuranose, a pH gradient would also be effective as long as the barriers of the transitions between the states are modulated by local pH, as they are by temperature.

As an interesting point: from a thermodynamic perspective, these results in a thermal gradient are necessary if the system has to transport heat from the warm to the cooler side (each part of the system must do it, not only the solvent). In the case of pH, likely the binding of protons around the molecule would change the barriers, and the fluxes would be used to transport protons from the regions where they are more abundant to the regions where they are less, as thermodynamics require.

We have added a sentence on the possible effects of pH gradients in our scenario in the final Discussion section.

8. This is blatant and I am not sure how it was missed between 8 authors, but the reference numbers are placed after punctuation, both after commas and full stops/periods. Reference numbers should be placed before the punctuation, within the sentence/fragment that they apply to.

As a matter of fact, curiously, this is the result of the automatic LaTeX formatting imposed by the ACS class style that we used to typeset our pre-print manuscript. Apparently, at ACS they like their punctuation before the superscript citations. It can be corrected manually, of course, and so we did in the revised manuscript. At any rate, the correct format will be imposed by the publishing team at NPG so as to comply with the style of Nat. Comm. if the paper is accepted.

As a funny interlude, here it is a screen capture of a few sentences from the introduction of a JACS paper, which demonstrate the curious ACS reference-number-after-punctuation style:

Self-propulsion of micro- and nanoscale objects can be achieved by harnessing the chemical free energy of the environment through substrate catalysis. For example, we and others have demonstrated that energy arising from catalytic reactions can drive the movement of asymmetric particles on the micrometer and submicrometer length scales by self-electrophoresis, self-diffusiophoresis, and bubble propulsion.¹ Autonomous motion of symmetric colloidal

Minor

1. Lines are not numbered, please do so next time.

We will certainly do.

2. There are problems with references, e.g. ref. nr 4. was published in PNAS and the journal name is another, strange one. Please double-check the reference list

We thank the referee for their meticulous reading. Indeed, that paper had the journal name wrong. We checked the references once more and, to the best of our resolving power, they all seem OK. There will be other checks if the paper is accepted when the proofs will be generated and cross-references to the cited papers will have to be produced.

3. Consider using "catalyst" instead of "catalyser"

We found one occurrence of the word “catalyser”, which has been replaced by “catalyst”.

4. grammar: "In this paper, we focus on a coarse-grained chemical reaction networks"

Corrected.

5. Figure 1, Figure 4- please use a) and b), instead of "left", "right" and "top", "bottom", placed accordingly in the figure, referenced properly in the text to e.g. Fig.1a instead of Fig. 1 (right panel)

Done *exactly* as instructed.

6. consider removing unnecessary statements as "It is not difficult to check that," and "It turns out that", they are distracting.

We have eliminated (the only) one occurrence of "It is not difficult to check that,"
We have eliminated (the only) one occurrence of "It turns out that,".

7. page 9, PH should read pH, velocit should ready velocity or speed

Corrected.

8. please remove "see again" when you refer to a figure for a second or third time. Just the figure number is sufficient, this is distracting.

All occurrences of “see again” have been replaced by “see”.

9. page 17, bottom "is of lower"

Corrected.

10. There are some weird commas (separating subject from verb) throughout the text, please double-check. E.g "The second requirement, amounts to"

Corrected. We double-checked and we could not spot other commas incorrectly separating subject from verb.

11. On my end, Figure S5, S6 and S8 (Top) do not show/the pdf is corrupted.

They look fine on our end. The production office will have the final word, if the paper is accepted for publication.

12. Consider digitizing the hand-drawn Figure 4 bottom.

In fact, the figure has been hand-drawn with an electronic stylus within a dedicated software. So the quality of the figure is comparable to that of a picture generated by any drawing software. Concerning the style, we quite like the hand-drawn feel when possible (also taking into account the ease and rapidity of production and the spectrum of possibilities inaccessible to the average user of specialized drawing software. Even though this is of course a matter of personal taste).

Reviewer #3

In this interesting work, the authors provided solid evidence that the nonequilibrium temperature gradient is crucial for the kinetic furanose selection in the ribose isomerisation network. I am enthusiastic about the publication of this work after addressing some comments below:

We thank the referee for their enthusiastic appreciation of our work. We have addressed their remarks as detailed below.

1. It is important to realize that the nonequilibrium states including the steady state are characterized by both the landscape and the temperature gradient which is very different from the equilibrium where the landscape barrier itself is enough. See Proc. Natl. Acad. Sci. USA , 105: 12271-12276. (2008); Advances in Physics, 64:1, 1-137. (2015); Reviews of Modern Physics 91(4), 045004 (2019).

The referee is absolutely right (we understand that what they mention as “landscape barrier” is indeed the free energy difference between two adjacent minima in the landscape). The interesting and timely review published in Rev. Mod. Phys. mentioned above has been added among the key references in the introduction.

2. The kinetic effect of the temperature gradient is a curl, or rotational flow driving the motion in state space as authors visioned as the cyclic motion. This has a rigorous mathematical foundation and can be quantified physically, as shown in Proc. Natl. Acad. Sci. USA , 105: 12271-12276. (2008); Advances in Physics, 64:1, 1-137. (2015); Reviews of Modern Physics 91(4), 045004 (2019).

The mathematical conceptualisation of non-equilibrium cycles offered by the recently developed landscape and flux theory is fascinating. We are definitely eager to study this formalism and apply it to our studies. We cited the PNAS and Advances in Physics papers suggested by the referee in the revised version when we discuss the characterization of nonequilibrium cycles in our system.

3. The effect of the nonequilibrium effect through the temperature gradient or the equivalent rotational driving force is to shift the fitness. There is a mathematical foundation behind this. See reference J. Chem. Phys., 137, 065102. (2012) to distinguish the evolution fitness and the population probability in the evolution. Under the equilibrium condition or Wright's studied simple cases, they are equal. However, when the nonequilibrium effects are important, the original fitness is no longer the optimal of the population any more.

The idea of nonequilibrium-induced shift of fitness on a given energy landscape is very interesting. We cite the 2012 JCP paper in the discussion section, when we talk about *selection of the fastest*. Once again, we shall investigate the nature of our nonequilibrium steady state and the ensuing selection in terms of the associated curl probability fluxes in the continuation of this and related projects.

REVIEWERS' COMMENTS

Reviewer #1 (Remarks to the Author):

Authors have well addressed my previous major comment. In particular, they have recognized that selection rules does not necessarily apply only to the starting conditions (abundance of a substrate), but also to its functional role in what will emerge later.

The manuscript can be accepted in the present form.